# Professional Involvement in Health Policy Development: A Cross-Sectional Study on Nurse Managers in Saudi Arabia

**DOI:** 10.3390/healthcare13222912

**Published:** 2025-11-14

**Authors:** Ahmad E. Aboshaiqah, Ahmed S. Alsadoun, Ahmad M. Rayani, Regie Buenafe Tumala

**Affiliations:** 1Administration and Education Department, College of Nursing, King Saud University, Riyadh 12375, Saudi Arabia; aaboshaiqah@ksu.edu.sa; 2Medical-Surgical Department, College of Nursing, King Saud University, Riyadh 12375, Saudi Arabia; ahalsadoun@ksu.edu.sa; 3Community and Psychiatric Mental Health Department, College of Nursing, King Saud University, Riyadh 12375, Saudi Arabia; arayani@ksu.edu.sa

**Keywords:** health policy, involvement, nursing, nurse leader, nurse manager, policy development, professional responsibility, Saudi Arabia

## Abstract

**Background/Objective:** Despite the vital role of nurses in health policy development, their involvement and influence in such development remain challenging. The involvement of nurse managers in health policy development in the Kingdom of Saudi Arabia (KSA) is not well examined. This study examined the extent of involvement of nurse managers in health policy development in the KSA and identified related barriers, benefits, confidence, and perceived impacts. **Methods:** A descriptive correlational cross-sectional design was used to achieve the study aim and objectives. An electronic self-administered questionnaire (Registered Nurses Involvement in Health Policy) was distributed to nurse managers working in the KSA. A total convenience sample of 238 nurse managers willingly and voluntarily agreed to participate. Data were collected from 10 February 2022 to 30 April 2022. Descriptive statistics and the Spearman rho correlation coefficient were used to analyze the data. **Results:** Among 238 nurse managers surveyed, 58% had received policy-related training, 73% reported high involvement as professionals, and 43% rated their confidence as high. Findings show the high level of involvement in health policy development of the nurse managers and their increased interest in influencing health policies. The policy activity most frequently chosen by the nurse managers is “provided health policy-related information to consumers or other professionals.” Moreover, the participants reported “lack of time, support, and resources” as their most perceived barrier and “improving public health” as their most perceived benefit. The majority of the nurse managers reported receiving information or training on health policies, and more than half of the participants rated their skills as “very good” or “excellent.” Our findings show the participants’ moderate level of confidence in performing health policy activities and moderate level of their perceived impact of their involvement in health policy activities on health outcomes. The results indicate a positive relationship between health outcomes and the ability of the nurse managers to influence health policy activities. **Conclusions:** The study findings suggest that Saudi nurse managers are increasingly engaged in policy development, but greater institutional support and targeted training are needed to strengthen their policy impact.

## 1. Introduction

Policies are intended to direct or influence the actions, behaviors, or decisions of others and reflect the policy developers’ values, principles, opinions, and beliefs (e.g., society, organizations, or individuals) [1]. A health policy refers to the decisions made by policymakers to promote individual health and can typically answer the questions of what, why, where who, when, and how regarding healthcare delivery [2]. Owing to the broad aspects of a health policy, it can affect the daily practice of healthcare providers.

Nurses comprise primary healthcare providers and represent nearly half of the global workforce and spend more time with patients than other healthcare professionals [3]. A nurse’s role extends beyond providing holistic care to providing care to patients and families. Nurses are a key stakeholder and leader in healthcare advancement. In addition, nurses have important responsibilities in healthcare research, education, and management and play a crucial role in health promotion, disease prevention, healthcare system support, global health, and professional development [3]. Such advantages can give nurses an opportunity to improve their position and role in health policy implementation and contribute to health policy development and planning [4,5,6,7]. The International Council of Nurses emphasizes the involvement of nurses in decision-making and health policy development to support efforts to prepare other nurses for health policy engagement [8].

Nurse managers are frontline leaders responsible for managing human and financial resources and maintaining a safe and satisfying environment for nurses, patients, and their families. Nurse managers are considered to be a liaison between nurses and the management by communicating and aligning the unit’s goals with the hospital’s strategic goals [9,10]. Furthermore, previous studies showed the flexibility of nurse managers’ role and ability to work during a crisis and under unusual circumstances, such as the COVID-19 pandemic [11,12]. The breadth of nurse managers’ skills, competencies, and practice can equip them with an extensive understanding of healthcare systems, which makes their involvement in health policy planning and development crucial for safe, accessible, and costly effective practice [3,6].

Nurses’ policy influence refers to their ability to affect healthcare affairs and decisions [7]. Despite the vital role of nurses in health policy development, their involvement and influence in such development remain challenging. A Delphi survey study that explored nurse leaders’ participation in health policy development in East Africa found that nurse leaders’ participation is limited and inconsistent [13]. Similarly, a study conducted in Kenya confirmed low involvement in health policy development. In addition, a limited number of nurse leaders were found to be involved in national health policy development [14]. Similar findings were reported in Nigeria and South Africa, where the involvement of nurses in health policy development and research is low and complex [15,16]. Likewise, the literature identifies challenges in nurses’ involvement in health policy development in the Middle East. A study conducted in Jordan on nurses’ involvement in health policy development demonstrated Jordanian nurses’ low involvement [17] but head nurses’ moderate involvement in health policy development [18]. Meanwhile, in Iran, the involvement of nurses in health policy development is moderate [19].

Many barriers prevent nurses’ involvement in health policy development. Al Faouri et al. [18] reported lack of time, support from others, and resources as the three most perceived barriers to Jordanian head nurses’ involvement in health policy development. Disappointment in work procedures and lack of external support, time, and access to key figures are the most perceived barriers of Iranian nurses [19]. Meanwhile, hierarchies and structural and nursing factors were reported as the primary barriers to nurses’ involvement in health policy in Kenya [14]. A recent systematic review identified the factors that can influence nurses’ involvement in health policy development as nursing-related factors (e.g., lack of priority on health policies, lack of knowledge in health policy development), management and organizational factors (e.g., lack of communication, access, and time), and work environment factors (e.g., lack of support from others and workload) [5].

In the Kingdom of Saudi Arabia (KSA), despite the ongoing healthcare transformation, the contribution of nurse managers remains underexplored. For instance, the KSA is experiencing a major national health sector transformation that aligns with the Saudi Vision 2030 strategic framework. The framework defined a roadmap toward the delivery of a value-based comprehensive, effective, and integrated healthcare system by considering economic needs and controlling healthcare expenditures [20,21]. As the largest group of health professionals in the KSA, nurses’ input will be crucial for the success of the transformation; thus, their leadership should be strengthened [22]. Hence, Alsufyani et al. [23] recommended aligning amendments for effective management in nursing policies and strategies with the transformation. However, the ability of the current nursing profession to meet the needs of the transformation is questionable [23]. The involvement of nurse managers in health policy development in the KSA is not well examined. The findings can contribute to nurses’ involvement in health policy domains, which is relatively low in the literature worldwide [24]. Furthermore, the findings can shed light on the involvement of nurses in management positions in health policy development in the KSA.

### Aim of the Study

This study explores how Saudi nurse managers engage in health policy development and identifies key barriers, benefits, and factors influencing their participation. Specifically, the study examines the level of involvement, perceived barriers and benefits, training sources, confidence, and perceived policy impact.

## 2. Materials and Methods

### 2.1. Research Design

A cross-sectional correlational descriptive research design was used to answer the aim and objectives of the study. Specifically, an electronic (online) self-administered questionnaire, utilizing Google Forms, was distributed to nurse managers working in the KSA.

### 2.2. Study Setting

Nurse managers working in primary, secondary, and tertiary healthcare settings in the KSA were conveniently targeted for this study. A total of 150 hospitals under the management of the MOH were targeted across the KSA.

### 2.3. Population and Sample

The target population was Saudi Arabian nurse managers currently working in different healthcare settings of the MOH in the KSA. In this study, a nurse manager is defined as a nurse in an administrative position with the decision-making power to affect the daily practice process. The definition includes nurses working in higher management, head nurses, supervisors, and charge nurses. The accessible population was nurse managers who the researchers had access to and received the electronic questionnaire. The rule of thumb formula was used to calculate the sample size, where the number of the variables in the study was × 10 + 50 [25]. This study has 13 variables of interest (six demographic and seven main variables); thus, a minimum sample size of 180 is required (13 × 10 + 50 = 180). This calculation was supported by using the G*Power version 3.1.9.7 software a priori: compute the required sample size—given effect size, alpha and power for test of correlation, revealing that a minimum computed sample size of 138 was adequate to achieve a medium effect size of 0.30, with a margin of error of 5% and a power of 95%.

In this study, convenience and networking sampling were conducted to recruit the participants. Nurse managers of Saudi citizenship and both genders were considered eligible to participate if they held a registered nurse license in the KSA and those who agreed to participate in the online survey were included. Chief nursing officers, nursing directors, nurse service managers, nurse supervisors, charge nurses, staff nurses, nursing aides, and healthcare assistants were not permitted to take part in the study’s online survey.

### 2.4. Study Instruments

An electronic self-administered survey designed by using Google Forms was conducted for this study. A reliable and valid questionnaire (i.e., Registered Nurses Involvement in Health Policy) [26] was employed as the research tool to assess the variables of interest. The tool was previously used in Middle Eastern studies [18,19]. The survey questionnaire consists of nine sections: (1) informed consent information; (2) demographics (age, education level, setting, bed capacity, years of experience, and department); (3) health policy activities, which encompass 13 activities related to health policies, rate of involvement as a professional and as a citizen, and level of interest in influencing health policies; (4) perceived barriers to health policy involvement; (5) perceived benefits of health policy involvement; (6) level of confidence in performing health policy activities; (7) perceived level of impact of health policy involvement on health outcomes; (8) level of skills related to health policy involvement; and (9) sources of knowledge and training.

The health policy development involvement of the nurse managers in the past two years was measured with the frequency of the health policy activities selected by the participants. In addition, the participants’ level of confidence in their ability to perform such activities and perceived impact of their health policy activity involvement on health outcomes were measured with two items rated on a Likert-type scale ranging from 1 (having no confidence or having no impact) to 5 (having confidence or having an impact). For level of confidence and perceived impact of health policy activity involvement, a median score of 4–5 was considered to be high, a median score of 3 was considered to be moderate, and a median score of 1–2 was considered to be low [18,26]. The cumulative confidence level score (13 activities) was calculated, and a cumulative score between 13 and 30, between 31 and 48, and between 49 and 65 was considered to be low, moderate, and high, respectively. Similarly, the cumulative score of perceived impact of health policy activity involvement (13 activities) was calculated, and a cumulative score between 13 and 30 was considered to be low, a cumulative score between 31 and 48 was considered to be moderate, and a cumulative score between 49 and 65 was considered to be high.

The rate of involvement in trying to influence health policies in the past two years as part of the professional responsibility of a registered nurse and as a citizen was assessed with a Likert-type scale ranging from 1 (not involved) and 5 (very involved). Similarly, the level of interest in trying to influence health policies was evaluated, and a median score of 4–5 was considered to be high, a median score of 3 was considered to be moderate, and a median score of 1–2 was considered to be low [18,26]. The perceived barriers to and benefits of involvement in health policy activities were measured by counting the frequency of the barriers and benefits selected by the participants. The participants were asked whether they received any information or training on health policy changes. Moreover, they were asked to identify their sources of information or training from a list of different sources. The participants were asked an additional question of whether they were aware of the existence of ethical policies regarding nurses’ involvement in health policy issues by Saudi health organizations (Saudi Commission for Health Specialties and Saudi Nursing Association). The participants were also instructed to rate their skills for impacting health policies obtained from their basic nursing education from “poor” to “excellent.”

In this study, a pilot study was conducted among 28 nurse managers who were excluded in the main study, to assess the reliability of the study tools by evaluating their internal consistency. The results of the reliability analysis revealed satisfactory Cronbach’s alpha values of 0.89, 0.87, 0.88, 0.93, and 0.95 for the following tools: involvement as a profession in health policy, involvement as a citizen in health policy, interest in health policy influence, level of confidence, and level of impact, respectively.

### 2.5. Ethical Considerations

This study was approved by an ethics committee. Specifically, ethical approval was obtained from the Central Institutional Review Board (IRB) of the Ministry of Health (MOH) in the KSA (IRB log No: 22-02 E). This study was conducted in accordance with the Declaration of Helsinki and good clinical practice guidelines. All the methods were used in accordance with relevant guidelines and regulations. Informed consent was obtained from all the participants. The instrument was distributed online, and the study information included the study title, the study purpose, the possible risks, confidentiality, the estimated time needed to answer the questionnaire, the voluntary participation of the nurses, and the participants’ right to withdraw from the study. The contact information of the principal investigator was provided in the study information sheet.

### 2.6. Data Collection Procedure

As this study obtained ethical approval from the central IRB of the MOH, it led the researchers to recruit nurse managers through their chief nursing officers or nursing directors, who are members of the Clinical Nursing Advisory Club (CNAC). The CNAC consists of chief nursing officers and nursing directors within the KSA. The online survey was distributed to eligible participants through an invitation sent to CNAC members. These members were encouraged to share the survey link with other chief nursing officers or nursing directors of the 150 hospitals managed by the MOH, who subsequently forwarded the survey link to their nurse managers that met the inclusion criteria. The distribution of study information and purpose was facilitated through CNAC members utilizing their group communications, such as WhatsApp group chats, and the targeted institutional email addresses of the participants. Data collection was conducted between 10 February 2022 and 30 April 2022.

### 2.7. Data Analysis

The participants’ demographic characteristics, level of involvement in health policy development, perceived barriers to and benefits of involvement in health policy activities, and health policy-related knowledge and skills were summarized by using descriptive statistics. The categorical variables were presented as numbers and percentages, and the continuous variables were presented as means and standard deviations (SDs) or medians. Spearman’s rho correlation coefficient was used to measure the relationships between the study variables, because the normality test conducted by using Kolmorgrov–Smirnov and Shapiro–Wilk tests indicated that the data were not normally distributed. A *p*-value of <0.05 is considered to be a significant level of confidence. All the calculations were performed by using SPSS 28.0 (SPSS Inc., Chicago, IL, USA).

## 3. Results

A total of 250 nurse managers participated in the online survey. Twelve responses had substantial missing data and therefore were excluded. Consequently, 238 responses were utilized in the final statistical analysis, resulting in a response rate of 95.0%. The participants’ characteristics, level of involvement in health policy development, perceived barriers to and benefits of involvement in health policy activities, health policy-related knowledge and skills, level of confidence in engaging in health policy activities, and perceived impact of health policy activity involvement on health outcomes are presented below.

### 3.1. Participants’ Characteristics

The mean age of the participants is 35.8 ± (SD = 6.7) years, with 138 (60%) of the participants belonging to the dominant age category of 30–39 years. The majority of the participants, that is, 128 (53.8%), has a bachelor’s degree, and 78 (32.8%) have a master’s degree. Moreover, 102 (42.9%) participants are working in a tertiary care setting, 72 (30.3%) are working in a secondary care setting, and 64 (26.9%) are working in a primary care setting. The majority of the participants, that is, 77 (33%), has work experience of 11–15 years, with a mean of 12.1 ± 6.8 years. Furthermore, the participants are working in different departments, such as medical/surgery (36 or 15.1%), nursing administration (33 or 13.9%), outpatient (29 or 12.2%), and pediatrics (27 or 11.3%). Details of the participants’ characteristics are presented in Table 1.

### 3.2. Participants’ Level of Involvement in Health Policy Development

The findings show that the mean score of the participants’ level of involvement in activities related to health policies as a professional is 3.5 ± 1.2 (median = 4), which indicates a high level of involvement. The mean score of the participants’ involvement in activities related to health policies as a citizen is 3.1 ± 1.2 (median = 3), which indicates a moderate level of involvement. The participants showed considerable interest in influencing health policies, with a mean of 3.9 ± 1.0 (median = 4). Table 2 presents the political activities of the participants in the past two years. The most frequently selected political activities of the participants are “provided health policy-related information to consumers or other professionals” (82 or 34.5%); “helped initiate or worked on a committee or coalition to take action on a health policy issue” (70 or 29.4%); “provided written reports and consultations, conducted research, or offered other types of assistance to a public official or for a health issue” (47 or 19.7%); “analyzed health policies and/or made recommendations about health policies to a public official” (46 or 19.3%); and “worked on a campaign for a candidate or health policy proposal” (43 or 18.1%).

### 3.3. Participants’ Perceived Barriers to and Benefits of Involvement in Health Policy Activities

Table 3 shows the barriers to involvement in health policy development, as perceived by the participants. Among the participants, 34 (14.3%) revealed that they perceived no barriers to their participation in health policy development. However, the participants’ most frequently chosen barriers are lack of time (102 or 42.9%), lack of support from others (89 or 37.4%), lack of money or other resources (53 or 22.3%), and employment restrictions (51 or 21.4%). Regarding the benefits of involvement in health policy activities, the most frequently chosen answers are improving the health of the public (135 or 56.7%), improving a situation or an issue (110 or 46.2%), opportunity to develop new skills (91 or 38.2%), ability to impact the health of many people at once (84 or 35.3%), and making a difference in others’ lives (81 or 34%). Among the participants, only 25 (10.5%) said that their involvement in health policy development would have no benefits. Table 4 presents the participants’ perceived benefits of their involvement in health policy development.

### 3.4. Participants’ Health Policy Knowledge and Skills

The majority of the participants reported receiving information or training on health policies (139 or 58.4%), and the source of such information or training was college coursework (83 or 34.9%), on-the-job experience (66 or 27.7%), workshops or conference sessions (65 or 27.3%), professional colleagues (51 or 21.4%), professional journals (43 or 18.1%), materials from professional organizations (36 or 15.1%), and mass media, such as TV, radio, and newspapers (32 or 13.4%). Among the participants, 104 (43.7%) were aware of the existence of ethical policies regarding nurses’ involvement in health policy issues by health organizations in the KSA. In comparison, 115 (48.3%) participants were unsure, and 19 (8%) were unaware of such policies. With regard to their skills related to health policy development, more than half of the participants rated their skills as “very good” (93 or 39.1%) or “excellent” (46 or 19.3%). However, 4 (1.7%) participants rated their skills as “poor,” 7 (7.1%) rated their skills as “fair,” and 78 (32.8%) rated such skills as “good.” The findings show a positive relationship between the participants’ level of skills and their level of involvement in health policy activities as a professional (r = 0.34, *p*-value ˂ 0.001) and as a citizen (r = 0.19. *p*-value 0.003). In addition, a positive relationship exists between the participants’ level of skills and their interest in trying to influence health policies (r = 0.50, *p*-value ˂ 0.001).

### 3.5. Participants’ Level of Confidence and Perceived Impact of Involvement in Health Policies

Our findings reveal a mean level of confidence of 42.2 ± 10.8 (median = 42) regarding the participants’ ability to engage in activities related to health policies, which indicates a moderate level of confidence. Meanwhile, the participants’ perceived impact of their health policy activity involvement on health outcomes is 45.7 ± 11.9 (median = 46), which indicates a moderate level of impact. A positive relationship exists between the participants’ level of confidence in their ability to perform health policy activities and their involvement in health policy activities as a professional (r = 0.31, *p*-value ˂ 0.001) and as a citizen (r = 0.32, *p*-value ˂ 0.001). Furthermore, a positive relationship exists between the participants’ perceived impact of their health policy activity involvement on health outcomes and their involvement in health policy activities as a professional (r = 0.27, *p*-value ˂ 0.001) and as a citizen (r = 0.27, *p*-value ˂ 0.001). Furthermore, a positive relationship exists between the participants’ level of confidence and perceived impact of their health policy activity involvement and their interest in trying to influence health policies (Table 5).

## 4. Discussion

The involvement and influence of nurse managers in health policy development remain challenging. This cross-sectional study investigates the extent of nurse managers’ involvement in health policy activities in the KSA. Moreover, this study reports the barriers to and the benefits of involvement in health policy activities, as perceived by the nurse managers. In addition, this study identifies the source of knowledge, skills level, confidence level, and perceived impact of health policy involvement on health outcomes of nurse managers in the KSA.

Unlike previous studies that reported low-to-moderate levels of staff nurses’ or nurse managers’ involvement in health policy activities [6,14,15,16,17,18,19], this study observes the sample’s high level of involvement as a professional nurse manager and moderate level of involvement as a citizen when measuring the score of the participants’ level of involvement in health policies. Furthermore, the participants demonstrated a high level of interest in influencing health policies. The discrepancy in the findings could be due to self-report or social desirability bias, as nurse managers tend to align their self-image with a more favorable view of their professional health policy engagement in comparison to their involvement as citizens. Al Faouri et al. [18] reported Jordanian head nurses’ moderate level of involvement in health policy activities as a professional and as a citizen. Moreover, the results indicated the head nurses’ moderate level of interest in influencing health policies [18]. In comparison, Jordanian registered nurses reported a low level of involvement in health policy development as a professional nurse and a moderate level of interest as a citizen [17]. Similarly, Barzegr et al. [19] found that Iranian nurses’ involvement in health policy development is moderate. Meanwhile, studies conducted in South Africa, East Africa, Kenya, and Nigeria indicated nurse leaders’ low or complex involvement in health policy activities [6,13,14,15,16]. Our findings reveal the nurse managers’ high level of involvement in health policy activities, which can be explained by the significant national health sector transformation as part of the Saudi Vision 2030 strategic framework [20,21]. As the largest group of health professionals, nurses’ contribution is crucial for the success of the transformation. However, the lack of studies in the KSA (for comparison before the transformation) makes confirming the conclusion difficult. In addition, the impact of the nurse managers’ involvement in health policy development is not clear and beyond the scope of this study. Nevertheless, the nurse managers’ high level of involvement and interest in health policy activities reflect the country’s positive health policy activity culture, which should be maintained and encouraged.

This study reveals that “provided health policy-related information to consumers or other professionals,” “helped initiate or worked on a committee or coalition to take action on a health policy issue,” and “provided written reports and consultations, conducted research, or offered other types of assistance to a public official or for a health issue” are the political activities most frequently chosen by the participants. The political activities align with those in Al Faouri et al. [18], which reported that the aforementioned political activities are among those most frequently chosen by Jordanian head nurses [18]. The findings are not surprising, because the political activities are part of nurses’ job responsibilities [18].

The study findings indicate that lack of time, support, and resources and employment restrictions are the most perceived barriers by the nurse managers. The results are similar to those of Al Faouri et al. [18], who found that the aforementioned barriers are the most frequently perceived by Jordanian head nurses. In addition, the barriers were identified in many other studies, as reported in a recent systematic review of studies with work environment and management and organization themes [5]. Nursing organizations stress improving nurses’ readiness for health policy development [8]. Understanding the factors that can hinder nurses’ participation in health policy activities is essential to improve their influential role in health policy activities [5]. Our findings reveal that the focus should be on strengthening organizational factors by reducing nurse managers’ workload and improving recruitment and environmental factors by providing them with appropriate resources and increasing their internal and external support for health policy activities.

Meanwhile, the most chosen benefits of health policy development involvement are improving the health of the public and a situation or an issue, providing an opportunity to develop new skills, having the ability to impact the health of many people at once, and making a difference in others’ lives. The benefits were also chosen by Jordanian head nurses [18]. The findings indicate that the nurse managers understand the core values and purpose of health policies to promote individual and societal health [2]. Al Faouri et al. [18] interpreted such findings as being rooted in the ethics and values of the nursing profession, which aims to improve the health of individuals and the community.

More than half of the participants rated their health policy-related skills from their primary nursing education as “very good” or “excellent.” Around 58% of the nurse managers indicated that they received information or training on health policies. College coursework, on-the-job experience, workshops, and conference sessions are the most cited sources of information or training received by the nurse managers. Public, organizational, and professional sources are also common health policy sources [27]. Using different health policy sources can increase nurses’ political competence and readiness to engage in health policy activities [27]. Improving their health policy abilities and skills is crucial for nurses to be rightfully involved in health policy development. Thus, nurse leaders and nursing organizations should provide access to a wide range of undergraduate, specialized graduate, and continuous professional education and training for health policy activities [15,24].

The findings reveal the nurse managers’ moderate level of confidence in performing health policy activities and perceived impact of their health policy activity involvement on health outcomes. In addition, the findings indicate a positive relationship between health outcomes and the nurse managers’ ability to influence health policy activity development as a nurse professional and as a citizen. Thus, increasing their involvement in health policy activities can enhance nurse managers’ confidence in implementing such activities and political activity experience. Ensuring influential nurses’ participation in health policy development was recommended by many previous studies [4,13,16,18]. Furthermore, encouraging nurses to effectively contribute to decision-making and health policy development should be included in strategic plans [24].

Overall, a recent umbrella review regarding nurses’ involvement in health policy development revealed that a lack of engagement stemmed from various multilevel barriers, hierarchical marginalization, and insufficient skills [28]. In connection, the results of the present study demonstrate significant relevance to recent evidence that indicates nursing complexity has a direct impact on both medical and organizational complexity [29]. This highlights the necessity of incorporating nursing data into health system planning [29]. The participation of nurse managers in policy-making would enhance their role within a more strategic, evidence-based policy framework, not only within the context of the KSA but also at a regional level in the Arab Gulf countries and on an international level. In particular, the study findings align with the Saudi Vision 2030 from multiple perspectives. For instance, the Saudi Vision 2030 aims to transform the healthcare system in the KSA by focusing on improvements in infrastructure, the adoption of digital health technologies, workforce empowerment, and the execution of innovative public health initiatives [22]. This vision aligns with the broader goals of economic diversification and societal reform, emphasizing the essential role of healthcare as a cornerstone for a thriving economy and a vibrant society [20]. In such economic transformation, Saudi nurses play a crucial role, with their contributions to chronic disease prevention and management highlighted in a previous systematic review [30]. Additionally, the vision addresses the challenges and opportunities that Saudi nursing faces, including workforce planning, task delegation, and community engagement, all of which are critical for the successful achievement of Vision 2030 [31]. The implications of this vision for the nursing profession are significant, as it aims to improve healthcare delivery and alleviate the ongoing shortage of qualified nursing professionals. By enhancing their engagement in health policy development and aligning their initiatives with Vision 2030, Saudi nurse managers can contribute to fostering a more efficient and high-quality care system.

### Limitations of the Study

This study is the first to investigate nurse managers’ involvement in health policy development in the KSA. However, the interpretation of this study’s findings should consider the study’s limitations. A major limitation of this study is its cross-sectional design that provided merely a snapshot of nurse managers during the study period, which may not truly represent changes over time and hindered the ability to establish causality. Another limitation is the use of convenience sampling, resulting in selection bias, as participants were selected based on their availability instead of being chosen through random selection. In addition, this study did not measure the impact of the nurse managers’ involvement in health policy development on the health system and health outcomes as well as their demographic characteristics. Consequently, these limitations impede the generalizability of the current study’s findings. Future studies should consider the limitations.

## 5. Conclusions

Nurse managers in the KSA reported a high level of involvement in health policy development, which reflects the country’s positive health policy activity culture. However, this study reveals the perceived barriers that can hinder nurse managers’ effective participation in health policy activities. Policymakers and nursing organizations in the KSA should address the barriers and support nurse managers to facilitate their involvement. Moreover, nurse managers’ level of confidence and perceived impact of their health policy activity involvement on health outcomes are positively correlated with their level of involvement in health policy development. Thus, nurse leaders and nursing organizations should provide access to a wide range of health policy development education and training to nurse managers in the KSA.

This study has implications for policymakers and professional and educational organizations. First, policymakers and practitioners should strengthen and encourage nurse leaders’ involvement in health policy development by considering the barriers to their health policy involvement and participation. Such encouragement will increase nurse managers’ confidence and health policy development experience. Furthermore, nurse managers’ health policy-related knowledge and skills can be improved by varying their information sources and improving their education. Finally, future studies should employ longitudinal or qualitative designs to explore causal relationships.

## Figures and Tables

**Table 1 healthcare-13-02912-t001:** Participants’ demographic characteristics.

Variable	Frequency n (%)
*Age* (missing = 8), mean ± SD	35.8 ± 6.7
20–29	36 (15.7)
30–39	138 (60)
40–49	48 (20.9)
≥50	8 (3.5)
*Level of education*	
Associate Degree	2 (0.8)
Diploma	26 (10.9)
Bachelor’s degree	128 (53.8)
Master’s degree	78 (32.8)
Doctorate (PhD) Degree	4 (1.7)
* Healthcare setting *	
Primary	64 (26.9)
Secondary	72 (30.3)
Tertiary	102 (42.9)
*Bed capacity*	
≤50	45 (18.9)
51–100	33 (13.9)
101–200	44 (18.5)
201–300	46 (19.3)
301–400	12 (5)
401–500	12 (5)
>500	46 (19.3)
*Years of experiences* (missing = 5), mean ± SD	12.1 ± 6.8
1–5	48 (20.6)
6–10	47 (20.2)
11–15	77 (33)
16–20	35 (15)
>20	26 (11.2)
*Departments/Units*	
Medical/Surgical	36 (15.1)
Operation Room	6 (2.5)
Oncology	6 (2.5)
Pediatrics	27 (11.3)
Intensive care	12 (5)
Emergency department	17 (7.1)
Dialysis	13 (5.5)
Obstetrics	7 (2.9)
Outpatient	29 (12.2)
Nursing Administration	33 (13.9)
Several Department	38 (16)
Others	14 (5.9)

Note. SD = Standard deviation; % = Percentage; PhD = Doctor of Philosophy.

**Table 2 healthcare-13-02912-t002:** The frequency of political activities that the participants have taken part in the past two years.

Activity	Frequency n (%)
Provided health policy-related information to consumers or other professionals	82 (34.5)
Helped initiate or worked on a committee or coalition to take action on a health policy issue	70 (29.4)
Provided written reports, consultations, research, or other assistance to a public official or for a health issue	47 (19.7)
Analyzed health policies and/or made recommendations about them to a public official	46 (19.3)
Worked on a campaign for a candidate or health policy proposal	43 (18.1)
Voted for a candidate or health policy proposal	39 (16.4)
Volunteered for a public official (not as part of a campaign)	31 (13)
Contacted a public official or their office regarding a health issue (such as calling, writing a letter, etc.)	29 (12.2)
Used mass media or public events to address a health policy issue	19 (8)
Lobbied, in person, a public policy-making body for a health policy-related issue	17 (7.1)
Drafted health policy legislation	14 (5.9)
Was an elected or appointed public official	14 (5.9)
Testified or did the research for a health-related legal action (lawsuit)	6 (2.5)
Others	3 (1.3)
None	6 (2.5)

Note. %—Percentage.

**Table 3 healthcare-13-02912-t003:** Perceived barriers to health policy involvement.

Activity	Frequency n (%)
Lack of time	102 (42.9)
Lack of support from others	89 (37.4)
Lack of money or other resources	53 (22.3)
Employment restrictions	51 (21.4)
Policymakers’ attitudes/values	41 (17.2)
Lack of access to key individuals	41 (17.2)
I have no barriers	34 (14.3)
Takes too long to see a difference	29 (12.2)
Frustration with the process	25 (10.5)
Others would not approve	24 (10.1)
Confronting others with opposing viewpoints	17 (7.1)
Do not know how to access or greater influence information	17 (7.1)
I have other personal priorities	15 (6.3)
Policy outcomes are too uncertain	14 (5.9)
My involvement would not make a difference	14 (5.9)
Policy activities do not make a difference	11 (4.6)
Do not see it as part of nursing care	10 (4.2)

Note. % = Percentage.

**Table 4 healthcare-13-02912-t004:** Perceived benefits of health policy involvement.

Activity	Frequency n (%)
Improving the health of the public	135 (56.7)
Improving a situation or issue	110 (46.2)
Opportunity to develop new skills	91 (38.2)
Ability to impact the health of many people at once	84 (35.3)
Making a difference in others’ lives	81 (34)
Professional duty (meeting standards of practice)	66 (27.7)
Being able to get involved/participate	59 (24.8)
Personal gratification	53 (22.3)
Personal Professionals create change by themselves advancement	48 (20.2)
Helps me to fulfill my personal agenda	44 (18.5)
Networking	41 (17.2)
Helping small groups of people who could not	37 (15.5)
Potential to obtain resources (funding, staffing)	27 (11.3)
I find no benefits	25 (10.5)

Note. % = Percentage.

**Table 5 healthcare-13-02912-t005:** Relationship between the level of confidence and the level of impact on the level of involvement and interest in health policy.

		Level of Confidence	Level of Impact
Involvement as a profession in health policy	Correlation coefficient	0.31	0.27
	Sig. (two-tailed)	˂0.001 ***	˂0.001 ***
Involvement as a citizen in health policy	Correlation coefficient	0.32	0.27
	Sig. (two-tailed)	˂0.001 ***	˂0.001 ***
Interest in health policy influence	Correlation coefficient	0.43	0.38
	Sig. (two-tailed)	˂0.001 ***	˂0.001 ***

Note. *** Significance level at 0.001.

## Data Availability

The raw data supporting the conclusions of this article will be made available by the authors on request due to legal reasons.

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
