# Peer review of "Professional Involvement in Health Policy Development: A Cross-Sectional Study on Nurse Managers in Saudi Arabia"

_healthcare, 2025, doi:10.3390/healthcare13222912_

Round 1

Reviewer 1 Report

Comments and Suggestions for Authors

Dear authors,

 The study was easy to read and to understand. I have some recommendations and questions. 

  • More information about the methods should be included in the abstract and also the sample size.
  • The study aims should be included before the methods section at the end of the introduction. 
  • Sampling was for convenience and should be indicated as a limitation. 
  • How were the managers contacted to give them the access to the google form?
  • The abstract says that you used Pearson but in the methods you say Spearman.
  • Did you calculate Cronbach alpha in your study or can you tell the adaptation results?
  • You say "15 variables of interest" but after that you say "9 demographic and 9 main variables)". Thus you should clarify it and also the sample size calculations. 
  • Please include what were the cut-off points for low, medium and high levels.
  • Table 5 should include a note with the analysis for p value. 
  • The discussion has citations in APA, please modify it to Healthcare citations. 

Kind regards

Author Response

Reviewer #1 Comments

The study was easy to read and to understand.

AUTHOR’S REPLY: Thank you very much, esteemed Reviewer #1 for this positive feedback.

I have some recommendations and questions. 

  • More information about the methods should be included in the abstract and also the sample size.

AUTHOR’S REPLY: Sample size and other relevant information were added. Please refer to Line/s 19-25.

  • The study aims should be included before the methods section at the end of the introduction. 
  • AUTHOR’S REPLY: The study aim and research questions have been specified before the methods section at the end of the introduction. Also, kindly know that this has been revised together with the feedback of the honorable Reviewer #2. Please refer to Line/s 133-137.

  • Sampling was for convenience and should be indicated as a limitation. 

AUTHOR’S REPLY: This has been added as one of the study’s limitations in Line/s 570-590.

  • How were the managers contacted to give them the access to the google form?

AUTHOR’S REPLY: This has been explained in more detail to the Data Collection Procedure section in the Methods. Please refer to Line/s 285-297.

  • The abstract says that you used Pearson but in the methods you say Spearman.

AUTHOR’S REPLY: We sincerely apologize for this typo error that should have been Spearman because the data are not normally distributed. This has been corrected in Line/s 24.

  • Did you calculate Cronbach alpha in your study or can you tell the adaptation results?

AUTHOR’S REPLY: The Cronbach alpha values have been added in Line/s 250-255.

  • You say "15 variables of interest" but after that you say "9 demographic and 9 main variables)". Thus you should clarify it and also the sample size calculations. 

AUTHOR’S REPLY: We sincerely apologize for this typo error because we failed to update this information from the proposal version. This has been corrected in Line/s 177-183.

  • Please include what were the cut-off points for low, medium and high levels.

AUTHOR’S REPLY: We respectfully clarify that cut-off points have been indicated in Line/s 212-249.

  • Table 5 should include a note with the analysis for p value. 

AUTHOR’S REPLY: The analysis for p value has been added as footnote.

  • The discussion has citations in APA, please modify it to Healthcare citations. 

AUTHOR’S REPLY: This has been modified throughout the Discussion section in Lines 441-568. Correspondingly, the references list has been corrected adhering to the Healthcare journal’s reference requirements.

Reviewer 2 Report

Comments and Suggestions for Authors

Abstract

Include key quantitative results , for example: “Among 238 nurse managers surveyed, 58% had received policy-related training, 73% reported high involvement as professionals, and 43% rated their confidence as high.”

Specify the objective in a clear sentence, for example: “This study examined the extent of nurse managers’ involvement in health policy development in Saudi Arabia and identified related barriers, benefits, confidence, and perceived impacts.”

Conclude with practical implications, for example: “These findings suggest that Saudi nurse managers are increasingly engaged in policy development, but greater institutional support and targeted training are needed to strengthen their policy impact.”

Introduction

Simplify the objective, for example: “This study explores how Saudi nurse managers engage in health policy development and identifies key barriers, benefits, and factors influencing their participation.”

Present it as a single concise sentence: “Specifically, the study examines the level of involvement, perceived barriers and benefits, training sources, confidence, and perceived policy impact.”

The transition from the global to the local context is abrupt. Add a linking sentence such as: “In Saudi Arabia, despite the ongoing healthcare transformation, the contribution of nurse managers remains underexplored…”

Methods

Clarify the sample size calculation. The text mentions a “rule of thumb formula” without justifying its use or the final number include an appropriate statistical reference or indicate whether power was calculated.

Detail inclusion and exclusion criteria. They are implied but not explicitly stated.

Improve consistency in statistical tests. The abstract mentions Pearson, while the text uses Spearman; it should be consistent (Spearman is correct since the data are non-normal).

Report the validity and reliability of the instrument, include reliability coefficients (e.g., Cronbach’s α).

Create clear subsections. Number and highlight subtitles such as “Design,” “Setting,” “Sample,” “Instrument,” and “Data Analysis.”

Results

The abstract mentions Pearson, but the text uses Spearman; this must be consistent (Spearman is correct since the data are not normally distributed).

Discussion

The section repeats the results instead of interpreting them. It starts by restating numbers and means already presented earlier.

The text mentions that the results are “similar to” or “different from” other studies but rarely explains why.

There is insufficient connection with the Saudi context although Vision 2030 is mentioned, it does not explore how this context might have contributed to the higher involvement observed.

The limitations are underdeveloped. They are mentioned superficially. It is necessary to explain their real impact (sampling bias, self-report bias, limited generalizability, etc.).

Conclusion

Include future recommendations, for example: “Future studies should employ longitudinal or qualitative designs to explore causal relationships.”

Avoid overgeneralizations, conclusions should be limited to the Saudi Arabian context and the study sample type.

Author Response

Reviewer #2 Comments

Abstract

Include key quantitative results , for example: “Among 238 nurse managers surveyed, 58% had received policy-related training, 73% reported high involvement as professionals, and 43% rated their confidence as high.”

AUTHOR’S REPLY: Thank you very much for this valuable suggestion. This has been added in Line/s 25-27.

Specify the objective in a clear sentence, for example: “This study examined the extent of nurse managers’ involvement in health policy development in Saudi Arabia and identified related barriers, benefits, confidence, and perceived impacts.”

AUTHOR’S REPLY: Thank you very much for this valuable suggestion. This has been added in Line/s 17-19.

Conclude with practical implications, for example: “These findings suggest that Saudi nurse managers are increasingly engaged in policy development, but greater institutional support and targeted training are needed to strengthen their policy impact.”

AUTHOR’S REPLY: Thank you very much for this valuable suggestion. This has been added in Line/s 40-47.

Introduction

Simplify the objective, for example: “This study explores how Saudi nurse managers engage in health policy development and identifies key barriers, benefits, and factors influencing their participation.”

AUTHOR’S REPLY: Thank you very much for this valuable suggestion. This has been added in Line/s 133-137.

Present it as a single concise sentence: “Specifically, the study examines the level of involvement, perceived barriers and benefits, training sources, confidence, and perceived policy impact.”

AUTHOR’S REPLY: Thank you very much for this valuable suggestion. This has been added in Line/s 133-137.

The transition from the global to the local context is abrupt. Add a linking sentence such as: “In Saudi Arabia, despite the ongoing healthcare transformation, the contribution of nurse managers remains underexplored…”

AUTHOR’S REPLY: Thank you very much for this valuable suggestion. This has been added in Line/s 113-114.

Methods

Clarify the sample size calculation. The text mentions a “rule of thumb formula” without justifying its use or the final number include an appropriate statistical reference or indicate whether power was calculated.

AUTHOR’S REPLY: This has been revised as suggested in Line/s 175-183.

Detail inclusion and exclusion criteria. They are implied but not explicitly stated.

AUTHOR’S REPLY: Details for inclusion and exclusion criteria have been added in Lines/s 184-192.

Improve consistency in statistical tests. The abstract mentions Pearson, while the t

ext uses Spearman; it should be consistent (Spearman is correct since the data are non-normal).

AUTHOR’S REPLY: We sincerely apologize for this typo error that should have been Spearman because the data are not normally distributed. This has been corrected in Line/s 24. Thank you very much for confirming the appropriate use of Spearman.

Report the validity and reliability of the instrument, include reliability coefficients (e.g., Cronbach’s α).

AUTHOR’S REPLY: The Cronbach alpha values have been added in Line/s 250-255.

Create clear subsections. Number and highlight subtitles such as “Design,” “Setting,” “Sample,” “Instrument,” and “Data Analysis.”

AUTHOR’S REPLY: This has been implemented throughout the Methods section as suggested. Please refer to Line/s 155-310.

Results

The abstract mentions Pearson, but the text uses Spearman; this must be consistent (Spearman is correct since the data are not normally distributed).

AUTHOR’S REPLY: Again, we sincerely apologize for this typo error that should have been Spearman because the data are not normally distributed. This has been corrected in Line/s 24. Thank you very much for confirming the appropriate use of Spearman.

Discussion

The section repeats the results instead of interpreting them. It starts by restating numbers and means already presented earlier. The text mentions that the results are “similar to” or “different from” other studies but rarely explains why. There is insufficient connection with the Saudi context although Vision 2030 is mentioned, it does not explore how this context might have contributed to the higher involvement observed.

AUTHOR’S REPLY: These have been addressed together in the revised and improved version of the Discussion section, especially last paragraph, as these are also pointed out by the honorable Reviewer #4.

The limitations are underdeveloped. They are mentioned superficially. It is necessary to explain their real impact (sampling bias, self-report bias, limited generalizability, etc.).

AUTHOR’S REPLY: This has been improved in Line/s 570-590. In addition, a separate sub-section as ‘Limitations of the Study’ has been added as suggested by the honorable Reviewer #3.

Conclusion

Include future recommendations, for example: “Future studies should employ longitudinal or qualitative designs to explore causal relationships.”

AUTHOR’S REPLY: This has been included in Line/s 610-611

Avoid overgeneralizations, conclusions should be limited to the Saudi Arabian context and the study sample type.

AUTHOR’S REPLY: This has been observed in the revised version of the Conclusion section in Lines/s 591-611.

Reviewer 3 Report

Comments and Suggestions for Authors

The article falls within the thematic scope of the journal; however, it requires answers to certain questions that seem crucial from a methodological point of view:

Lines 116–120:
How were the nurses selected? What criteria were used? Were the questionnaires sent to all nurse managers working in KSA, or only to a selected group? The research design is unclear. It requires clarification and elaboration.

Lines 126–127:
What percentage of nurse managers who received the questionnaires ultimately participated in the study? How many nurse managers working in Saudi Arabia could have potentially participated in the study? Was participation limited to only 250 nurse managers – that is, the number to whom the questionnaires were sent – or was it possible for more to participate (lines 206–207)?
Answers to these questions are important, as they help determine whether the study presented in the article is representative of Saudi Arabia.

Lines 122–124:
There is no information on the following: What was the time period during which the study was conducted?

Suggestion:
The section on limitations (lines 337–388) could be separated from the Discussion and presented under a separate heading (as Limitations). This would improve the clarity of the article.

Author Response

Reviewer #3 Comments

The article falls within the thematic scope of the journal; however, it requires answers to certain questions that seem crucial from a methodological point of view:

AUTHOR’S REPLY: Thank you so much for your valuable feedback, honorable Reviewer #3. Below is our point-by-point response to each of your comments for improvement.

Lines 116–120:
How were the nurses selected? What criteria were used? Were the questionnaires sent to all nurse managers working in KSA, or only to a selected group? The research design is unclear. It requires clarification and elaboration.

AUTHOR’S REPLY: This has been clarified and elaborated in Line/s 156-160.

Lines 126–127:
What percentage of nurse managers who received the questionnaires ultimately participated in the study? How many nurse managers working in Saudi Arabia could have potentially participated in the study? Was participation limited to only 250 nurse managers – that is, the number to whom the questionnaires were sent – or was it possible for more to participate (lines 206–207)?
Answers to these questions are important, as they help determine whether the study presented in the article is representative of Saudi Arabia.

AUTHOR’S REPLY: The answers to the above questions are indicated in Line/s 285-297.

Lines 122–124:
There is no information on the following: What was the time period during which the study was conducted?

AUTHOR’S REPLY: This has been added in the Data Collection Procedure section. Please refer to Line/s 297 and also in the Abstract.

Suggestion:
The section on limitations (lines 337–388) could be separated from the Discussion and presented under a separate heading (as Limitations). This would improve the clarity of the article.

AUTHOR’S REPLY: This has been separated as Limitations of the Study section. Please refer to Line/s 570-590.

Reviewer 4 Report

Comments and Suggestions for Authors

The proposed topic is relevant, timely, and aligned with global efforts to enhance nurses’ engagement in health policy. The paper provides valuable descriptive data and contributes to a growing body of literature on nursing leadership and policy participation. However, before this work can be considered for publication, several methodological, structural, and formal aspects should be carefully revised:

1) The abstract is overly descriptive and does not clearly express the research gap, methods, and core findings. Please condense the background and emphasize why this study was needed and what it adds beyond existing Middle Eastern evidence. The results could be more focused on key numeric findings (mean involvement scores, main barriers, and relationships) rather than listing all details. The conclusion should highlight implications for nursing policy and leadership development rather than restating results.

2) While the design (cross-sectional, correlational) is appropriate, several issues require clarification. The paper mentions a “rule of thumb” for sample size but applies a non-probability convenience/network sampling, which limits generalizability. Please clarify this inconsistency and discuss its impact on external validity. How was the response rate of 95% obtained in an online survey? Were incomplete or duplicate responses excluded? Specify whether the tool was culturally adapted and validated for Saudi participants (translation, pilot testing, Cronbach’s α). This is essential for psychometric credibility. The authors report use of both Pearson and Spearman correlations. Given that normality was violated, why was Pearson mentioned earlier? Please ensure consistency in the methods section.

3) The finding of a “high level of involvement” appears inconsistent with the moderate level of confidence and moderate perceived impact. Could this discrepancy be due to self-report or social desirability bias? Please discuss. The results are primarily descriptive; adding inferential comparisons (e.g., involvement by education level, years of experience, or healthcare setting) would improve analytical depth.

4) Some tables are long and repetitive (especially Tables 2–5). Consider summarizing or merging where possible.

5) The discussion reiterates many results but offers limited interpretation. Strengthen this section by connecting your findings to broader theoretical or conceptual frameworks, for example, “policy influence competence,” “political skill,” or “nursing advocacy capacity.” Please, consider: how do organizational factors (e.g., workload, role conflict) affect nurse managers’ ability to engage in policy-making? Could gender or managerial hierarchy influence access to policy processes in the Saudi context?

6) Overall, the discussion section could be further strengthened by elaborating on how nurse managers’ engagement in policy activities can impact the broader organization of care and the equitable distribution of resources. When addressing the need to “reduce workload” and “strengthen organizational factors,” the authors could link these managerial priorities to the development of policies that formally recognize and measure nursing care complexity within hospital performance and financing frameworks. Recent evidence (PMID: 39857934) highlights that nursing complexity directly influences both medical and organizational complexity, underscoring the importance of integrating nursing data into health system planning. Framing nurse managers’ policy involvement in this broader systems perspective would enhance the depth of the discussion and situate their contribution within a more strategic, evidence-based policy context.

7) The reference list does not follow MDPI style (missing DOI, inconsistent capitalization, use of “et al.” inappropriately). Please reformat according to Healthcare guidelines. Ensure each in-text citation corresponds precisely to the reference list. Some are missing page numbers or incorrect initials. Verify hyperlinks (e.g., WHO, ICN, Vision 2030) are current and properly formatted. The paper would benefit from 1–2 additional recent international references (2023–2025) on nurses’ policy engagement to update the literature base.

8) The manuscript is understandable but would benefit from language polishing for conciseness and fluency. For instance, avoid redundancy (“health policy development and policy-making process”). You should simplify long sentences for clarity and ensure uniform tense usage (past tense for results, present for general statements).

Comments on the Quality of English Language

The manuscript is understandable but would benefit from language polishing for conciseness and fluency. For instance, avoid redundancy (“health policy development and policy-making process”). You should simplify long sentences for clarity and ensure uniform tense usage (past tense for results, present for general statements).

Author Response

Reviewer #4 Comments

The proposed topic is relevant, timely, and aligned with global efforts to enhance nurses’ engagement in health policy. The paper provides valuable descriptive data and contributes to a growing body of literature on nursing leadership and policy participation. However, before this work can be considered for publication, several methodological, structural, and formal aspects should be carefully revised:

AUTHOR’S REPLY: We highly appreciate your valuable comments, honorable Reviewer #4. We earnestly hope that our revisions are satisfactory and worthy of consideration for the publication of the revised and improved version of our work. Thank you so much and Godspeed!

1) The abstract is overly descriptive and does not clearly express the research gap, methods, and core findings. Please condense the background and emphasize why this study was needed and what it adds beyond existing Middle Eastern evidence. The results could be more focused on key numeric findings (mean involvement scores, main barriers, and relationships) rather than listing all details. The conclusion should highlight implications for nursing policy and leadership development rather than restating results.

AUTHOR’S REPLY: This has been revised as suggested in Line/s 13-47.

2) While the design (cross-sectional, correlational) is appropriate, several issues require clarification. The paper mentions a “rule of thumb” for sample size but applies a non-probability convenience/network sampling, which limits generalizability. Please clarify this inconsistency and discuss its impact on external validity. How was the response rate of 95% obtained in an online survey? Were incomplete or duplicate responses excluded? Specify whether the tool was culturally adapted and validated for Saudi participants (translation, pilot testing, Cronbach’s α). This is essential for psychometric credibility. The authors report use of both Pearson and Spearman correlations. Given that normality was violated, why was Pearson mentioned earlier? Please ensure consistency in the methods section.

AUTHOR’S REPLY:

As support to the use of ‘rule of thumb’ in sample size calculation, we added using the G*Power version 3.1.9.7 software for correlation test through A priori: compute the required sample size – given effect size, alpha and power. Hence, we respectfully request the expert guidance of the honorable Reviewer #4 regarding this. Please refer to Line/s 175-183. Thank you so much.

We also added Cronbach’s alpha values based on the reliability results from the pilot study. Please refer to Line/s 250-255.

In addition, the missing data from 12 responses have been indicated in Line/s 314-315.

We sincerely apologize for this typo error that should have been Spearman because the data are not normally distributed. This has been corrected in Line/s 24.

3) The finding of a “high level of involvement” appears inconsistent with the moderate level of confidence and moderate perceived impact. Could this discrepancy be due to self-report or social desirability bias? Please discuss. The results are primarily descriptive; adding inferential comparisons (e.g., involvement by education level, years of experience, or healthcare setting) would improve analytical depth.

AUTHOR’S REPLY: The discrepancy of results has been pointed out and provided with explanation in Line/s 456-459. For adding inferential comparisons on involvement by education level, years of experience, or healthcare setting, we humbly and respectfully request the honorable Reviewer #4 to exclude this because it is not included in our study’ aim and objectives as well as due to the limited time. Kindly know that we are open and more than willing to make additional changes for the improvement of our work. Thank you very much and Godspeed!

4) Some tables are long and repetitive (especially Tables 2–5). Consider summarizing or merging where possible.

AUTHOR’S REPLY: We adjusted the tables to fit one page.

5) The discussion reiterates many results but offers limited interpretation. Strengthen this section by connecting your findings to broader theoretical or conceptual frameworks, for example, “policy influence competence,” “political skill,” or “nursing advocacy capacity.” Please, consider: how do organizational factors (e.g., workload, role conflict) affect nurse managers’ ability to engage in policy-making? Could gender or managerial hierarchy influence access to policy processes in the Saudi context?

AUTHOR’S REPLY: This has been addressed in the revised and improved version of the Discussion section in Line/s 441-568, particularly in the last paragraph.

6) Overall, the discussion section could be further strengthened by elaborating on how nurse managers’ engagement in policy activities can impact the broader organization of care and the equitable distribution of resources. When addressing the need to “reduce workload” and “strengthen organizational factors,” the authors could link these managerial priorities to the development of policies that formally recognize and measure nursing care complexity within hospital performance and financing frameworks. Recent evidence (PMID: 39857934) highlights that nursing complexity directly influences both medical and organizational complexity, underscoring the importance of integrating nursing data into health system planning. Framing nurse managers’ policy involvement in this broader systems perspective would enhance the depth of the discussion and situate their contribution within a more strategic, evidence-based policy context.

AUTHOR’S REPLY: AUTHOR’S REPLY: This has been addressed in the revised and improved version of the Discussion section in Line/s 441-568, particularly in the last paragraph.

7) The reference list does not follow MDPI style (missing DOI, inconsistent capitalization, use of “et al.” inappropriately). Please reformat according to Healthcare guidelines. Ensure each in-text citation corresponds precisely to the reference list. Some are missing page numbers or incorrect initials. Verify hyperlinks (e.g., WHO, ICN, Vision 2030) are current and properly formatted. The paper would benefit from 1–2 additional recent international references (2023–2025) on nurses’ policy engagement to update the literature base.

AUTHOR’S REPLY: This has been completely addressed based on the journal’s requirements regarding citation and referencing format. Also, four references have been added as suggested, to update the literature base.

8) The manuscript is understandable but would benefit from language polishing for conciseness and fluency. For instance, avoid redundancy (“health policy development and policy-making process”). You should simplify long sentences for clarity and ensure uniform tense usage (past tense for results, present for general statements).

AUTHOR’S REPLY: Kindly know that we submitted the revised version of our work for English language editing.

Round 2

Reviewer 2 Report

Comments and Suggestions for Authors

I consider that the changes made by the authors make the article suitable for publication.

Reviewer 3 Report

Comments and Suggestions for Authors

I have no more comments.

Reviewer 4 Report

Comments and Suggestions for Authors

Congratulations to the authors for the improvements.